# The Evaluation of *CYP2D6, CYP2C9, CYP2C19*, and *CYP2B6* Phenoconversion in Post-Mortem Casework: The Challenge of Forensic Toxicogenetics

**DOI:** 10.3390/metabo13050661

**Published:** 2023-05-16

**Authors:** Arianna Giorgetti, Sara Amurri, Giulia Fazio, Carla Bini, Laura Anniballi, Filippo Pirani, Guido Pelletti, Susi Pelotti

**Affiliations:** Unit of Legal Medicine, Department of Medical and Surgical Sciences, University of Bologna, Via Irnerio 49, 40126 Bologna, Italy

**Keywords:** drug metabolism, drug–gene interaction, cytochrome P450 genes, genotype, phenotype, phenoconversion

## Abstract

In toxicogenetics, an integrative approach including the prediction of phenotype based on post-mortem genotyping of drug-metabolising enzymes might help explain the cause of death (CoD) and manner of death (MoD). The use of concomitant drugs, however, might lead to phenoconversion, a mismatch between the phenotype based on the genotype and the metabolic profile actually observed after phenoconversion. The aim of our study was to evaluate the phenoconversion of *CYP2D6, CYP2C9, CYP2C19*, and *CYP2B6* drug-metabolising enzymes in a series of autopsy cases tested positive for drugs that are substrates, inducers, or inhibitors of these enzymes. Our results showed a high rate of phenoconversion for all enzymes and a statistically significant higher frequency of poor and intermediate metabolisers for *CYP2D6, CYP2C9*, and *CYP2C19* after phenoconversion. No association was found between phenotypes and CoD or MoD, suggesting that, although phenoconversion might be useful for a forensic toxicogenetics approach, more research is needed to overcome the challenges arising from the post-mortem setting.

## 1. Introduction

The human cytochrome P450 (CYP) drug-metabolising enzymes (DMEs), such as *CYP2D6, CYP2C9, CYP2C19*, and *CYP2B6*, are subject to genetic polymorphism, which gives rise to individual genotypes with the correspondent genotype-predicted phenotypes classified in poor (gPM), intermediate (gIM), normal (gNM), and ultrarapid metabolisers (gUM) [1,2,3,4]. This inter-genotypic variability accounts for the variability in drug response, particularly drug efficacy and/or safety [5]. Indeed, based on drugs activity, gIM and gPM might have higher plasma concentrations with an increased risk for toxicity; on the other hand, in gUM, the excessive metabolism can put subjects at risk for therapeutic failure [6]. Several studies have focused on the prediction of the functional phenotype associated with these discrete genotype groups, aiming to develop a “personalised medicine” as the best approach for the choice of “the right drug at the right dose the first time” [2,7].

To translate genotypic data into a DME phenotypic prediction, an activity score (AS) system has been proposed for *CYP2D6* and has gained acceptance among the scientific community [8]. The score has also been adopted by the Clinical Pharmacogenetics Implementation Consortium and by other organisations, which have developed guidelines on the genotype-specific doses for a number of drugs [9]. 

It is known that, beyond genetic factors, drug metabolism can be also influenced by non-genetic factors such as sex, age, weight, physiological conditions or diseases, diet, and use of concomitant drugs. These factors account for the phenomenon of phenoconversion, that is, a mismatch between the genotype-based prediction of cytochrome P450-mediated drug metabolism and the real metabolising capacity [7,10,11]. The genotype–phenotype discrepancy might explain some inconsistencies that have been found within genotype-based association studies. Particularly, the use of concomitant medication might trigger drug–drug and drug–drug–gene interactions (DDGI), especially when taking drugs with inhibitory or inductive effects on DME [10,12], e.g., the co-consumption of paroxetine, a strong *CYP2D6* inhibitor, and tamoxifen, a drug metabolised by the same DME [12,13]. 

*CYP2D6* and *CYP2C19* are the most characterised enzymes for phenoconversion because they are highly involved in the metabolism of the main prescribed drugs and the most polymorphic ones [5,14]. Although phenoconversion is a transient phenomenon, it could be particularly relevant in the populations of genotypic IM, so the actual number of subjects with PM phenotype could be much higher than that identified with the only genotype-based phenotype. An integrative approach taking into consideration DNA-based phenotype and phenoconversion due to DDGI might help to prevent adverse drug reactions (ADR), which represent one of the top ten leading causes of death in developed countries [15].

In the forensic scenario, the medico-legal death investigation including genetic testing is known as “molecular autopsy” and could be particularly useful for sudden unexpected deaths or drug-related deaths [16,17,18]. Indeed, the toxicogenetics approach could avoid misinterpretation of toxicological results in evaluating the cause of death (CoD) and the manner of death (MoD) [6]. The aim of this study was to assess the magnitude of phenoconversion of *CYP2D6, CYP2C9, CYP2C19*, and *CYP2B6* enzymes in casework of deceased subjects tested positive for drugs and psychoactive substances in order to evaluate the role of toxicogenetics in the forensic setting. 

## 2. Materials and Methods

### 2.1. Study Population, Inclusion Criteria, and Sample Collection

Thirty-five caseworks were included in the study. Inclusion criteria were as follows: a. cases submitted to a complete autopsy at the Department of Medical and Surgical Sciences, University of Bologna between 2020 and 2022; b. autopsies which had been submitted to a systematic toxicological analysis and tested positive for drugs and psychoactive substances metabolised by, or inhibitors/inducers of, *CYP2D6, CYP2B6, CYP2C9*, and *CYP2C19*; c. post-mortem interval defined by autopsy as <5 days. Exclusion criteria included undetermined CoD, unsuccessful isolation of DNA, or limited volume for further analyses. During the autopsy, samples of urine, bile, peripheral (femoral) blood or, in the absence of peripheral blood, aortic or heart blood, as well as gastric content were collected for toxicological analysis. An additional blood sample was collected for DNA analysis. The blood specimens were preserved with 2% sodium fluoride. All specimens were stored at −20 °C immediately following collection at autopsy.

A database of anonymised post-mortem data was built containing age, gender, ethnicity, and medical history, with particular focus on drug use disorders, other psychiatric diseases, neurologic diseases, or infectious diseases connected to drug use, e.g., HIV or hepatitis C, CoD and MoD, toxicological results, and genetic analysis. 

CoD were classified as follows: 14.3% mono-intoxications, 54.3% mixed intoxications, and 31.4% non-intoxications. MoD were classified as accidental in 74.3% of cases, suicidal in 17.1%, and natural in 8.6%.

### 2.2. Systematic Toxicological Analysis

For each case, a comprehensive toxicological analysis, including general screening and quantification of drugs of abuse and medicinal drugs, was performed. Particularly, analyses for alcohol were done by gas chromatography coupled to a Flame Ionization Detector (Shimadzu QP 2010 Plus, Kyoto, Japan). Blood samples were initially screened for illicit drugs (cannabinoids, cocaine, opiates, methadone, and amphetamines/methamphetamines/MDMA/MDA) by immunoassay (ILab 650, Werfen, Barcelona, Spain) [19]. Confirmation analyses for cannabinoids were performed with a Shimadzu GC-2010 Plus gas chromatograph equipped with a model AOC-6000 auto-sampler system and interfaced with a QP 2010 Ultra mass spectrometer (Shimadzu, Kyoto, Japan) using a previously validated method [20]. Confirmation analyses for other illicit drugs and screening/confirmation for 68 psychoactive medications (benzodiazepines, Z-drugs, antipsychotics, antidepressants, and medical opioids) were performed with an ACQUITY UPLC^®^ System (Waters Corporation, Milford, MA, USA) equipped with an Acquity UPLC^®^ HSS C18 column (2.1 × 150 mm, 1.8 μm; Waters) using a previously validated method [21,22]. 

### 2.3. CYP2D6, CYP2C9, CYP2C19, and CYP2B6 Genotyping

In order to provide the genotype assessment for *CYP2D6*, *CYP2C9*, *CYP2C19,* and *CYP2B6*, 200 μl of whole blood from each sample was extracted using the QIAamp^®^ DNA Mini and Blood Mini (Qiagen, Hilden, DE, USA) following the DNA Purification from Blood or Body Fluids (Spin Protocol) protocol. Each DNA extract was eluted in 50 μl of buffer ATE. Quality and quantity of extracted DNA were determined using the Quantifiler™ Trio DNA Quantification Kit (Applied Biosystems, Foster City, CA, USA) according to the manufacturer’s recommended protocol. SNP genotyping for detection of the analysed *CYP2D6* most common allelic variants (Appendix A) was performed with the TaqMan^®^ SNP Genotyping Assay (Applied Biosystems, USA) using the QuantStudio5 Real-Time PCR System. The ready-to-order TaqMan^®^ SNP Genotyping Assay for *CYP2D6* genotype is provided by Thermo Fisher Scientific Inc. (Waltham, MA, USA) and the reaction mixture consisted of 5 µL of TaqPath ProAmp Master Mix, 0.5 µL of TaqMan^®^ SNP Genotyping Assay 20× Primer Mix (*2: C__27102425_50 and C__27102414_10, *3: C__32407232_L0, *4: C__27102431_D0, *6: C__32407243_20, *9: C__32407229_60, *10: C__11484460_40, *17: C___2222771_A0, *41: C__34816116_20, *59: C__72649938_20), and 5 ng of genome DNA template in a total volume of 10 µL. For genotyping analysis, the following PCR conditions were used: pre-read step at 60 °C for 30 s, an initial denaturation and enzyme activation step at 95 °C for 5 min followed by 40 cycles at 95 °C for 15 s and at 60 °C for 60 s, and a post-read step at 60 °C for 30 s. For each run samples, known genotypes were included as positive controls as well as negative control.

*CYP2D6* genotyping data were analysed using the Thermo Fisher Cloud Genotyping Application [23]. The software collects raw data from genotyping experiments and represents allelic attribution using a scatter plot. The auto-call method in the GT application module of Genotyping automatically generates allele discrimination plots with well-separated clusters for genotype callings and the call rate. To investigate the number of *CYP2D6* gene’s copies, a TaqMan CopyNumber Variant assay (Thermo Fisher Scientific Inc. Waltham, MA, USA) was performed using the QuantStudio™ 5 Real-Time PCR System. The reaction mix composition was 1.1 µL of TaqMan^®^ Copy Number Assay (assay id: Hs00010001_cn [Ex9]), 1.1 µL of TaqMan^®^ Copy Number Reference Assay with an internal control TERT-Human (assay id: 4403316), 11 µL of TaqPath ProAmp Master Mix, and 20 ng of DNA template. For copy number variant analysis, the PCR conditions were as follows: an initial denaturation and enzyme activation step at 95 °C for 10 min followed by 40 cycles at 95 °C for 15 s and at 60 °C for 60 s. Samples with a known number of copies were used as positive controls, and negative controls were also included in each run. Relative quantification was performed using CopyCaller™ Software v.2.1 (Thermo Fisher Scientific Inc., Waltham, MA, USA) following the comparative ΔΔCt method. The software returns histograms showing the number of copies of the *CYP2D6* gene for each sample examined. 

The *CYP2C9, CYP2C19,* and *CYP2B6* allelic variants (Appendix A) were analysed by the SNaPshot minisequencing method, as previously described by Carano et al. [24]. No duplication and deletion assays were performed for *CYP2C9, CYP2C19,* and *CYP2B6* genes due to their low frequency in the reference population [25,26].

### 2.4. Activity Score, Phenotype, and Phenoconversion Assessment

Software PHASE v.2.1.1 [27] was used to infer the gametic phase and to assign the most probable haplotype to every subject included in this study from SNP data. Then, diplotype (from here on referred to as “genotype”) was inferred following the Pharmacogene Variation (PharmVar) Consortium database [28]. For samples where software calculated two possible combinations of haplotypes, only those with greater frequency were considered. In order to assess the phenotype based on genotype for *CYP2D6* and *CYP2C9*, an activity score (AS) was attributed. The AS allows to assign, for each allele, a value of 0 for null, 0.5 for intermediate, 1.0 for wild-type, and two times these scores for the corresponding gene duplication genotypes. The AS, which is a numerical variable, is then used to assign the phenotype category, enabling the prediction of the individual metabolising capacity [29]. The attribution of AS was based on tables available on the PHARMGKB website [30,31], which follows Clinical Pharmacogenetics Implementation Consortium (CPIC) and Dutch Pharmacogenetics Working Group (DPWG) guidelines [32,33,34]. 

To translate the AS into phenotype, the contiguous consensus scale proposed by Caudle et al. was followed [31]. This model has been developed for *CYP2D6* and predicts that AS = 0 values correspond to phenotypic group PM, AS values 0 < x < 1.25 to IM, AS values 1.25 ≤ x ≤ 2.25 to NM, and AS values > 2.25 result in categorization into the UM group. For *CYP2B6* and *CYP2C19*, the AS method could not be applied, thus, the attribution to phenotypes was based on tables available on PHARMGKB, and consensus standardized terms for phenotypes were used [34,35]. Considering the method used by Hicks et al. for *CYP2D6* [1] to assign the activity prediction based on genotype test, we propose the use of “g-phenotype”, which includes gPM, gIM, gNM, and gUM, for all the analysed CYPs.

Once the metabolic phenotype of the subjects was obtained, the role of inhibitor or inducer drugs detected by toxicological analysis was assessed for the phenoconversion of the corresponding enzymes. Following the suggestion of Shah et al. [7], we used the term “p-phenotype” to distinguish the phenotypic categories based on the true genotype (g-phenotype) from their phenotypic counterparts resulting from phenoconversion (p-phenotype). The evaluation of inhibitors and inducers of *CYP2D6, CYP2C9, CYP2C19*, and *CYP2B6* was performed using the full version of Cytochromes P450 Drug Interaction Table [11,30,33,34,35,36,37,38,39,40,41]. To evaluate the p-phenotypes of *CYP2D6*, AS values were adjusted based on the application of the rules described by Borges et al. [29], which introduced the use of inhibition factors (e.g., multiplication of the AS by 0 in case of a strong inhibitor, by 0.5 in case of a weak inhibitor). Consequently, the predicted *CYP2D6* phenotype was adjusted following Cicali et al. [13]. For *CYP2C9* and *CYP2C19*, a protocol for the adjustment of the AS value was not available, therefore the metabolic profiles have been adjusted as reported by Mostafa et al. [36]. *CYP2B6* adjustment of the metabolic profile in the presence of phenoconversion was performed as reported by Mangò et al. [42]. In this model, CYP2B6 activity (p-phenotype) was classified in the following categories: PM, low IM-PM, high IM, high IM-extensive metaboliser (EM), NM, and EM. In order to obtain a more homogeneous classification, g- and p-phenotype UM were included into EM. In Appendix A, the parameters used for defining the g-and p-phenotypes are reported. In individuals where both inducer and inhibitor drugs were detected, the g-phenotype of the four DMEs was not phenoconverted due to the limited evidence and the absence of a consensus to guide the conversion [43].

### 2.5. Data and Statistical Analyses

Descriptive statistics was provided for all data. An ANOVA (Analysis of Variance) test, assuming a Gaussian distribution of the age, was used to test the difference in age between men and women. For the quantification of drugs of abuse, mean and standard deviation (SD) of the blood levels were calculated. After inclusion, all cases were separated based on the CoD into three groups: a. mono-intoxications, when only one substance was retrieved as CoD; b. mixed intoxications, when multiple substances contributed to the toxicity; c. non-intoxications, when CoD was any other cause but a fatal intoxication. Associations between categorical variables (gender, ethnicity, CoD, or MoD) were assessed by Chi square analysis. For all the associations between categorical variables, e.g., to test whether phenoconversion (g-phenotype vs. p-phenotype) led to a difference in the frequency of PM, IM, NM, and UM, the Chi square test was applied. The Chi square test was also used to explore the distribution of p-phenotype in the three CoD groups and in the MoD groups. Statistic tests were performed with Stata 15.1 (StataCorp LLC, College Station, TX, USA) and were considered significant with *p*-values < 0.05. The figures were realized with Prism (GraphPad Software, LLC, version 9.0.0).

## 3. Results

In the 35 cases included in the study, mean age was 43.5 years (standard deviation: 14.3), ranging from 25 to 85 years. When classifying the age into the 4 groups, 34.3% of subjects were aged 18–35 years, 37.1% were aged 36–50 years, 20% were aged 51–65 years, and 8.6% were aged >65 years. Seven deceased individuals were women, 20% of the total. Men and women did not statistically differ for mean age, as checked by ANOVA test. The majority of the deceased were Europeans (85.7%), while 5 subjects (14.3%) were considered from the Near Eastern ancestry group [30,44], particularly, 3 subjects came from the Mediterranean region (8.6%) and 2 were from the Arabian Peninsula (5.7%). Past history included alcohol or drug use disorder in 13 (37.1%) cases, psychiatric diseases in 12 cases (34.3%), infectious diseases in 3 cases (8.6%, including hepatitis C and HIV), and epilepsy or other neurological diseases in 3 cases (8.6%), and these conditions were often combined. A negative clinical history was observed in 7 cases (20%), and no clinical data were available for 5 subjects (14.2%). 

Among classical drugs of abuse, methadone was detected in 19 cases (54.3%), with a mean level of 599.2 ng/mL (SD 483.8), ranging from 33 to 1679 ng/mL. Nine subjects tested positive and were quantified for morphine (25.7%), with a mean concentration of 423.3 ng/mL (SD: 330.8), ranging from 19 to 1069 ng/mL. Cocaine was detected in 7 cases (20%), with a mean level of 2860 ng/mL (SD: 2583), ranging from 137 to 6393 ng/mL. Regarding medicinal drugs, 20 subjects tested positive for co-consumed benzodiazepines, 20 for antidepressant drugs, 15 for antipsychotics, 8 for opioid medications (e.g., tramadol, fentanyl, dextromethorphan, hydromorphone), 4 for paracetamol, 2 cases included other neuroleptics, particularly chlorpromazine and valproic acid, and 6 other substances.

In Appendix A, details of age, gender, ethnicity, past history, CoD and MoD, and results of toxicological analyses are presented. 

### 3.1. CYP2D6, CYP2C9, CYP2C19, and CYP2B6 Genotype and g-Phenotype

All the samples were successfully genotyped for *CYP2D6* and *CYP2C9.* For *CYP2B6,* 34 samples were genotyped, and only 28 samples for *CYP2C19* due to the high degradation index of 7 samples. In Table 1 are shown the genotype frequencies and the corresponding g-phenotypes of all the samples. One subject (2.86%) classified as *CYP2D6* gPM showed a no function allele (**4*) together with the deletion of the other allele (**5*), one subject (2.86%) classified as *CYP2C9* gPM showed a no function allele (**3*) and a reduced function allele (**2*), the *CYP2C19* gPM (3.57%) showed two no function alleles (**2*), and no gPM was found for *CYP2B6*. The frequencies of *CYP2D6, CYP2C9, CYP2C19*, and *CYP2B6* gIMs were 31.43%, 31.43%, 21.43%, and 41.18%, respectively. For NMs g-phenotype, there were 17 subjects (48.57%) for *CYP2D6*, 23 subjects (65.71%) for CYP2C9, 21 subjects (75%) for CYP2C19, and 13 subjects (38.23%) for *CYP2B6*. The 6 subjects (17.14%) classified as *CYP2D6* gUMs harboured **1/*1* ×3, **1/*2* ×3, **1/*41* ×4, **2/*2* ×3, **2/*2* ×4 gene duplications. The 7 subjects (20.59%) classified as *CYP2B6* gEMs showed allelic variants with increased activity *(*4,*22*).

### 3.2. Phenoconversion, Activity Score (AS) Adjustment, and Statistical Analyses

Toxicological analysis showed the following substances with inhibitory actions against *CYP2D6*: amiodarone, citalopram, clozapine, levomepromazine, methadone, sertraline, trazodone, venlafaxine, and 11-hydroxy-Δ9-tetrahydrocannabinol (11-OH-THC); Δ9-tetrahydrocannabinol (Δ9-THC) as a moderate/weak inhibitor [41]; and chlorpromazine, cocaine, haloperidol, paroxetine, and fluoxetine as strong inhibitors [37]. For chlorpromazine, cocaine, and haloperidol, only in vitro data were available. Olanzapine, metoprolol, and lidocaine were found in 5 samples, and literature data regarding their impact on *CYP2D6* activity, whether weak or strong inhibitors, are lacking. In this study, we opted to take a conservative safety-oriented approach, considering them as weak inhibitors. According to the literature, *CYP2D6* does not have inducers [45]. 

Substances with inhibitory actions against *CYP2C9* were amiodarone, olanzapine, paroxetine, and valproic acid, all classified as weak or moderate inhibitors. The inducers detected were phenobarbital and warfarin. 

Amiodarone, citalopram, diazepam, fluoxetine, olanzapine, nordiazepam, valproic acid, and warfarin, all classified as weak or moderate inhibitors of *CYP2C19*, were detected, and also the two inducers of the same enzyme: carbamazepine and phenobarbital. 

Finally, toxicological analysis showed the following substances with inhibitory actions on *CYP2B6*: paroxetine, sertraline, and 11-OH-THC. Diazepam (only in vitro) and methadone were found as inducers [46,47,48]. In Figure 1, the distribution of g-phenotypes and the corresponding p-phenotypes for the analyzed CYP enzymes are reported. Results of p-phenotype after phenoconversion referring to the inducers and inhibitors are shown in Table 2. The case-by-case description of g-phenotype and p-phenotype is reported in Appendix A.

When evaluating the association between g-phenotype categories (gPM, gIM, gNM, and gUM) and p-phenotype, a statistically significant result was obtained for *CYP2D6*, *CYP2C9*, and *CYP2C19* (*p* < 0.001), showing a higher frequency of PMs and IMs over NMs and UMs when considering the p-phenotype. The test could not be applied to the *CYP2B6* enzyme due to the different classification applied for g-phenotype and p-phenotype. The distribution of p-phenotype within the CoD and MoD groups for all the evaluated enzymes showed no statistically significant association between variables (*p* > 0.05).

## 4. Discussion

In the forensic practice, in order to understand CoD and MoD, toxicological results have to be considered in a comprehensive evaluation together with circumstantial, clinical, autoptic, and histological data. In recent years, several studies have focused on the utility of post-mortem genotyping for pharmacogenes as a complementary analysis aiding the interpretation of toxicological results [4] in a toxicogenetics approach. In the present study, 35 deceased individuals tested positive for substances metabolised by, or inducers/inhibitors of *CYP2D6*, *CYP2C9, CYP2C19*, and *CYP2B6* were typed for the corresponding gene polymorphism. On the basis of the genotyping and of the activity score-based assessment, the phenotypic categories gPM, gIM, gNM, and gUM were identified, and we adopted the term g-phenotype to group and to distinguish them from the p-phenotype, the term used to identify the phenotype after phenoconversion. The majority of individuals were gNMs for *CYP2D6* (48.57%), CYP2C9 (65.71%), and CYP2C19 (75%), but for *CYP2B6*, gNMs (38.23%) and gIMs (41.18%) were almost equally represented. The g-phenotype distribution in our sample, which was mostly represented by European individuals, showed for *CYP2D6* a greater number of gUM (17.14%) compared to the observed distribution in the Italian population [24]. For *CYP2B6*, the highest rate of gEMs could be due to the 20.6% of subjects in our population from the Near Eastern groups [49]. Nevertheless, due to the limited number of our samples, the statistical analysis was not performed, but the findings would require further investigations on a larger sample size, which is difficult to collect in the forensic setting. On the basis of the post-mortem genotyping, however, the majority of the included cases would seem to display in general a not so compromised metabolising capacity for the studied DMEs and would not be considered at high risk for drugs toxicity. However, the risk of ADRs is not only affected by the phenotype based on the genotype but also by phenoconversion, especially in subgroups of the population with a high rate of this phenomenon, which could lead to negative outcomes [7].

The main aim of the present work, indeed, consisted of the evaluation of the magnitude of the phenoconversion phenomenon in a forensic setting where the co-consumption of multiple drugs metabolised or acting as inhibitors/inducers on *CYP2D6*, *CYP2B6*, *CYP2C9*, and *CYP2C19* enzymes was observed. To the best of our knowledge, in the post-mortem setting, the phenoconversion has been evaluated only in a single case of acute intoxication involving venlafaxine, showing a mismatch between the pharmacogenetics testing and the phenotype evaluation on the basis of the metabolic ratio [50]. The authors reported that the phenoconversion was the likely phenomenon explaining this discrepancy [50]. 

In the assessment of phenoconversion, a first step is represented by the evaluation of the possible influence of drugs on the selected DMEs. A first challenge in our study arose considering that the online databases reporting the role of drugs as inducer/inhibitor of CYP enzymes are restricted only to a certain number of drugs. Moreover, in our casework, both inhibitors and inducers were simultaneously detected and, according to the literature, the evaluation of the p-phenotype should not be performed in this scenario due to limited scientific evidence [43]. However, *CYP2D6* and, to a lower extent, CYP2C19 and *CYP2C9* have been widely studied for phenoconversion in the clinical setting, so multiple guides exist to assess the phenomenon [29,33,51]. 

Considering the *CYP2D6* gene, our results showed an alarming rate of phenoconversion from a class of genotype-based gNM to p-phenotyped IM or PM (15 out of 17 gNMs, 88.2%), mostly due to co-consumption of strong inhibitors such as paroxetine and fluoxetine [7]. Methadone is also a weak inhibitor, and, in our casuistry, this substance was particularly consumed together with antidepressants or CNS depressant drugs, which are substrates for *CYP2D6*. This might lead to unexpected severe reactions due to the co-consumed therapeutic drugs and to a possible increase of their role in the methadone toxicity. Individuals with gIM genotype are considered more likely to be susceptible to phenoconversion due to their intrinsically already compromised capacity to mediate drug metabolism [7]. In our casuistry, 4 out of 11 (36.4%) of the g-phenotype gIMs were classified as p-phenotype PMs, but even higher rates of phenoconversion were seen for g-phenotype gUMs. Indeed, five out of six (83.3%) individuals turned into p-phenotype PM or IM. Given the fact that only three subjects remained UMs or NMs (8.6%) after the evaluation of phenoconversion, this phenomenon might be problematic in terms of toxicity and more relevant for forensic toxicogenetics.

It must, however, be highlighted that the strength of some inhibitors, whether strong or weak, e.g., lidocaine and olanzapine, is controversial in the literature, and that for some drugs, particularly cocaine, which is frequently detected in post-mortem acute intoxications, only in vitro data are available [37]. The categorization of the strength of inhibition, whether weak or strong, is another challenge highlighted by the present work and should be considered for future post-mortem studies.

A high rate of phenoconversion was also shown for *CYP2C19*, since more than half of the cases originally g-phenotyped as gNMs (13 out of 21 cases, 61.9%) were classified as PMs on the basis of the p-phenotype. The same happened to g-phenotyped gIMs, given that three out of six subjects had PM p-phenotypes. Although primarily metabolised by *CYP2B6*, methadone is also a substrate for *CYP2C19*, and the inhibition due to other drugs should be considered in the post-mortem setting. Other relevant substrates of *CYP2C19* are represented by paroxetine and quetiapine, both CNS depressant drugs, the role of which might be enhanced in the case of co-consumption of inhibitors such as diazepam, citalopram, and olanzapine.

*CYP2C9* is more rarely studied for phenoconversion with respect to *CYP2D6* and *CYP2C19*. Our results, by indicating that 7 out of 32 subjects might have been phenoconverted from gNMs to PMs, seem to encourage this type of analysis. Indeed, substrates of *CYP2C9* are represented by several BDZ, e.g., diazepam and temazepam, commonly identified in our cases, as well as by some opioids, e.g., dextromethorphan and hydromorphone, which might be relevant in the determination of the CoD [52].

*CYP2B6* represents a minority of the hepatic CYP protein content [42] but is associated with significant interindividual variations in pharmacokinetics of several drugs, including methadone and antidepressant drugs [53,54,55], which are particularly relevant in forensic toxicology. The concept of phenoconversion has been rarely applied to this enzyme but is recommended to be considered during *CYP2B6* phenotype prediction [42,49,56]. Benzodiazepines such as diazepam and midazolam seem to induce transcriptional expression of the *CYP2B6* gene and, in liver tissue, were associated with increased enzyme activity [42]. In our study, the co-consumption of diazepam, methadone, and other *CYP2B6* inducers appeared to ameliorate the function of *CYP2B6* predicted from genotype in 16 out of 31 cases (51.6%), while this effect was rare for other *CYP450* enzymes. On the other hand, a reduction in the estimated function of *CYP2B6* was seen in 10 cases (32.3%) changing from a g-phenotype gEM to a p-phenotype high-IM or from a g-phenotype gIM to a p-phenotype PM, mainly due to the co-consumption of paroxetine. Considering the high rate of subjects that tested positive for methadone and the high rate of possible modifications of its pharmacokinetics due to phenoconversion, this phenomenon seems extremely relevant and should be taken into account, particularly in the case of methadone-related intoxications. On the other hand, the challenges faced in the translation of genotype into g-phenotype and in the evaluation of the p-phenotype highlight the need for more consistent and standardized procedures in order to take into account non-genetic factors in the prediction of patients’ drug-metabolising capacity [42,45].

In our study, statistical analyses confirmed that the frequency of metabolic classes (gPM, gIM, gNM, and gUM) of *CYP2D6, CYP2B6, CYP2C9*, and *CYP2C19* were significantly influenced by the assessment of the phenoconversion, with a higher representation of PMs or IMs among p-phenotypes compared to g-phenotypes. 

In forensic toxicogenetics, the p-phenotype might be associated with the CoD or MoD, and, particularly, PMs and IMs might be more represented in mono- and mixed intoxications compared to non-intoxications. However, for our samples, the association between p-phenotypes and CoD or MoD was not found, so CoD and MoD were only based on the post-mortem examination and toxicology results. On the other hand, other possible relevant factors for phenoconversion, such as age and gender, were also not associated with CoD and MoD, suggesting that a higher sample size might be needed in order to evaluate the significance of all the factors involved in phenoconversion. Indeed, in the present study, the number of included cases is one of the main drawbacks, although it is roughly in line with past studies focused on the forensic setting [16]. Moreover, to the best of the authors’ knowledge, this is the first attempt to assess phenoconversion in the post-mortem setting. In addition, our inclusion criteria involved post-mortem cases submitted to a judicial autopsy with post-mortem intervals of <5 days. This criterion was set in order to avoid longer post-mortem intervals, which are known to affect the quality of the available DNA in the whole blood collected at autopsies [49,56]. However, this might have reduced the number of included cases. On the other hand, the storage conditions were not always available, and, despite this criterion, some cases yielded too low DNA amounts to proceed with the genetic analyses, limiting the sample size.

Considering the genetic analyses, phase II enzymes and drug transporters, which are well known to play a role in drug absorption, disposition, toxicity, and efficacy, were not considered in the present study. However, phase II metabolising enzymes seem less important than phase I [6], and limited data are available for assessing phenoconversion, which was the focus of our study. As another limitation of the present study, our toxicological analysis did not include a quantification method covering the full spectrum of metabolites of the detected parent drugs, so a parent/metabolite ratio could not be achieved. This would require the development and validation of a specific method, which is currently ongoing and might be used for future research. Finally, in this study, only co-consumed drugs were considered for phenoconversion, while there is evidence that other factors, e.g., gender, age, and pathological conditions, especially impacting on the liver [57], have a considerable impact on the activity of CYP enzymes. 

It has to be highlighted that the real phenotype cannot be measured with certainty in the post-mortem setting, but only estimated on the basis of the rules applied in the clinical setting by previous authors. This was particularly controversial for *CYP2B6* given the fact that phenoconversion for this DME has been more rarely investigated. Despite the limitations and the challenges faced in the present work, a high rate of possible phenoconversion was demonstrated, highlighting the fact that DDGIs should not only be considered when scribing patients with polypharmacology or evaluating them in the clinical setting, but should be included within the set of analyses performed post-mortem. Indeed, forensic toxicogenetics testing could be profitably applied in routine forensic casework and brought into the courtroom, but more scientific studies are needed [58]. Nevertheless, in forensic toxicogenetics, to understand the actual role of phenoconversion in the evaluation of the cause of death, data that are not always available at the post-mortem examination, such as the dose of the drug taken, the method of administration, and the survival time, would be necessary. 

## 5. Conclusions

Forensic toxicogenetics is facing ever more cases of multi-drug consumption, involving therapeutic drugs as well as drugs of abuse, often coupled to long-term psychotropic use history. The evaluation of phenoconversion remains a difficult task due to the lack of experimental and casework-related data in the post-mortem setting. Despite several challenges, our study demonstrated a high rate of phenoconversion due to drug–drug–gene interactions. A complete genotype/phenotype/phenoconversion evaluation might be useful in order to better evaluate analytical results in the definition of the CoD and MoD, and future research should be devoted to this scope on wider casework and a larger panel of drug-metabolising enzymes. Furthermore, future studies should address the need for a systematic and standardized approach to research in the post-mortem setting. 

## Figures and Tables

**Figure 1 metabolites-13-00661-f001:**
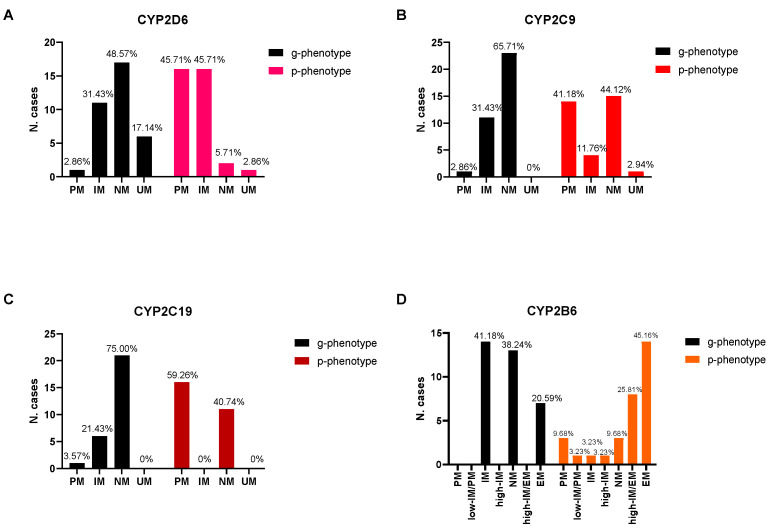
g-phenotype and p-phenotype distributions for *CYP2D6* (**A**), *CYP2C9* (**B**), *CYP2C19* (**C**), and *CYP2B6* (**D**). PM: poor metabolisers, IM: intermediate metabolisers, low IM-PM: low intermediate-poor metabolisers, high IM: high intermediate metabolisers, high IM-EM: high intermediate-extensive metabolisers, NM: normal metabolisers, EM: extensive metabolisers, and UM: ultrarapid metabolisers.

**Table 1 metabolites-13-00661-t001:** *CYP2D6*, *CYP2C9*, *CYP2C19,* and *CYP2B6* genotype frequencies and the corresponding g-phenotype.

Genotype	n.	%	g-Phenotype
*CYP2D6*			
**1/*1*	7	20.00	gNM
**1/*1* ×3	2	5.71	gUM
**1/*2*	4	11.43	gNM
**1/*2* ×4	1	2.86	gUM
**1/*4*	5	14.29	gIM
**1/*41* ×3	1	2.86	gUM
**2/*2* ×3	1	2.86	gUM
**2/*2* ×4	1	2.86	gUM
**2/*2*	4	11.43	gNM
**2/*4*	2	5.71	gIM
**2/*41*	2	5.71	gNM
**4/*41*	1	2.86	gIM
**10/*10*	1	2.86	gIM
**41/*41*	1	2.86	gIM
**4/*10*	1	2.86	gIM
**4/*5*	1	2.86	gPM
Total	35	100%	-
*CYP2C9*			
**1/*1*	23	65.71	gNM
**1/*2*	9	25.71	gIM
**1/*3*	1	2.86	gIM
**2/*2*	1	2.86	gIM
**2/*3*	1	2.86	gPM
	35	100%	-
*CYP2C19*			
**1/*1*	21	75	gNM
**1/*2*	6	21.43	gIM
**2/*2*	1	3.57	gPM
Total	28	100%	-
*CYP2B6*			
**1/*1*	7	20.59	gNM
**1/*4*	6	17.65	gEM
**1/*5*	5	14.71	gNM
**1/*6*	10	29.41	gIM
**1/*7*	3	8.82	gIM
**1/*9*	1	2.94	gIM
**1/*22*	1	2.94	gEM
**2/*5*	1	2.94	gNM
Total	34	100%	-

**Table 2 metabolites-13-00661-t002:** Phenoconversion process for *CYP2D6, CYP2C9, CYP2C19*, and *CYP2B6.* Samples were divided into different g-phenotypic groups (PM, IM, NM, UM, EM). Starting AS: activity score based on genotype. Adjusted AS: activity score adjusted according to the models described in the literature. IM: intermediate metabolisers, NM: normal metabolisers, PM: poor metabolisers, UM: ultrarapid metabolisers, EM: extensive metabolisers. g-phenotype: phenotype based on genotype, P-phenotype: phenoconversion-induced phenotype. N: number of samples divided by found drugs, inhibitors (weak, moderate, or strong), and inducers.

Starting AS	g-Phenotype (n)	Inhibitors in Blood		Inducers in Blood	Adjusted AS	p-Phenotype
*CYP2D6* (n = 35)						
		Effect (n)	Drug(s) Detected	Drug(s) Detected		
0	gPM (1)	Strong (0)	-	-	-	-
		Weak (1)	Citalopram	-	0	PM
		None (0)	-	-	-	-
0 < x < 1.25	gIM (11)	Strong (4)	Chlorpromazine * Cocaine * Haloperidol * Paroxetine	-	0	PM
		Moderate/Weak (7)	Amiodarone Clozapine Levomepromazine Methadone Sertraline Trazodone 11-OH-THC	-	0.25 < x < 0.5	IM
		None (0)	-	-	-	-
1.25 ≤ x ≤2.25	gNM (17)	Strong (7)	Cocaine * Fluoxetine Paroxetine	-	0	PM
		Moderate/Weak (8)	Citalopram Levomepromazine Lidocaine ** Methadone Olanzapine ** Trazodone Venlafaxine 11-OH -THC	-	0.75 < x < 1	IM
		None (2)	-	-	-	NM
>2.25	gUM (6)	Strong (4)	Chlorpromazine * Citalopram Cocaine * Paroxetine	-	0	PM
		Moderate/Weak (1)	Methadone Trazodone	-	1.5	NM
		None (1)	-	-	-	UM
*CYP2C9* (n = 35)						
		Effect (n)	Drug(s) Detected	Drug(s) Detected		
0–0.5	gPM (1)	Strong/Moderate/Weak (1)	Paroxetine	-	-	PM
1–1.5	gIM (11)	Strong/Moderate/Weak (6)	Paroxetine Δ9-THC	-	-	PM
		Strong/Moderate/Weak (1)	Valproic acid	Phenobarbital	-	n.d.
		None (4)	-	-	-	IM
2	gNM (23)	Strong/Moderate/Weak (7)	Amiodarone Olanzapine Paroxetine Sertraline Δ9-THC Valproic acid	-	-	PM
		None (1)	-	Warfarin		UM
		None (15)	-	-	-	NM
*CYP2C19* (n = 28)						
		Effect (n)	Drug(s) Detected	Drug(s) Detected		
	gPM (1)	None (1)	-	-		
-	gIM (6)	Strong/Moderate/Weak (3)	Amitriptyline Diazepam Sertraline	-	-	PM
		None (3)	-	-	-	IM
-	gNM (21)	Strong/Moderate/Weak (13)	Amiodarone Citalopram Diazepam Fluoxetine Nordazepam Olanzapine Δ9-THC Valproic acid Warfarin	-	-	PM
-		Strong/Moderate/Weak (1)	Diazepam Valproic acid	Phenobarbital	-	n.d.
-		None (7)	-	-	-	NM
*CYP2B6* (n = 34)						
		Effect (n)	Drug(s) Detected	Drug(s) Detected		
-	gIM (14)	Strong/Moderate/Weak (3)	Paroxetine	-	-	PM
-		None (8)	-	Diazepam * Methadone	-	high-IM\EM
-		Strong/Moderate/Weak (2)	11-OH-THC Paroxetine	Diazepam * Methadone	-	n.d.
-		None (1)	-	-	-	IM
-	gNM (13)	None (8)	-	Diazepam * Methadone	-	EM
-		Strong/Moderate/Weak (1)	Paroxetine	Diazepam * Methadone	-	n.d.
-		Strong/Moderate/Weak (1)	Paroxetine	-	-	Low-IM/PM
-		None (3)	-	-	-	NM
-	gEM (7)	Strong/Moderate/Weak (1)	Paroxetine	-	-	High-IM
-		None (3)	-	Diazepam * Methadone	-	EM
-		None (3)	-	-	-	EM

Samples showing both weak and strong inhibitors were counted among those with strong inhibitors. * = in vitro only data available. ** = data in the literature are lacking about their impact on *CYP2D6* activity. They are considered as weak inhibitors. n.d. = not determined.

## Data Availability

Data is contained within the article or Appendix A.

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
