# Peer review of "The Evaluation of CYP2D6, CYP2C9, CYP2C19, and CYP2B6 Phenoconversion in Post-Mortem Casework: The Challenge of Forensic Toxicogenetics"

_metabolites, 2023, doi:10.3390/metabo13050661_

Round 1
Reviewer 1 Report
The manuscript titled “The evaluation of CYP2D6, CYP2C9, CYP2C19 and CYP2B6 phenoconversion in post-mortem casework: the challenge of forensic toxicogenetics” attempted to examine the genotype-based phenotype in post-mortem cases and the extent of phenoconversion in those samples.
As such the work in the manuscript is interesting. However, there are some minor deficiencies in the manuscript which need to be addressed to make it publishable. There are several short paragraphs and, in some cases, only one sentence in a paragraph. Instead of several short paragraphs, try to consolidate them based on the idea/concept. This is applicable to all sections including the introduction and discussion. On page 2, line 74, CYP450 should be CYP as that’s the standard acronym and has also been used in this manuscript in other instances. The supplementary materials are useful for the readers.
Author Response
We would like to thank the reviewer for his precious comment and for giving us the opportunity to revise our work. We appreciated his constructive comments on our manuscript, which led to a much-improved thorough revision of the manuscript. Our detailed answers – in bold - are interleaved to the reviews below.
Corrections and amendments made are also marked in yellow in the revised version of the manuscript.
R: “… There are several short paragraphs and, in some cases, only one sentence in a paragraph. Instead of several short paragraphs, try to consolidate them based on the idea/concept. This is applicable to all sections including the introduction and discussion. On page 2, line 74, CYP450 should be CYP as that’s the standard acronym and has also been used in this manuscript in other instances. … “
A: Many thanks for the comments, which allowed us to improve the quality of the paper. The manuscript has been extensively rewritten in order to better clarify the aim and the concept of the study, also removing short paragraphs throughout the manuscript. We have also consolidated the short subparagraphs (by removing subparagraphs 2.3.1, 2.3.2 and 2.3.3, 2.4.1, 2.4.2 and 2.4.3, as well as 3.2) and corrected page 2, line 74, which has been rephrased.
Reviewer 2 Report
This study focuses on the evaluation of specific CYP450 enzymes, however the aim is not easy to understand. The authors should highlight mainly in the abstract and better in the introduction and in other parts the purpose of the study.
In general the whole manuscript needs careful proof reading. Some sections in Methods include Tables which could be moved in supporting info, parts 2.3 and 2.4 should be carefully written in a comprehensive manner. Results and discussion parts are not appropriate, a bit chaotic, without a specific row and must be reorganized. Perhaps the authors could present some specific targets for future studies.
The terms "g-phenotype" and "p-phenotype" should be explained better providing specific bibliography otherwise should be corrected.
Conclusion section seems like an opinion. I wonder if authors would address some major points and challenges of some demanding questions of the discussed area.
Detailed references should be provided for genotyped variants CYP2D6, CYP2C9, CYP2C19 and CYP2B6.
Primers for RT-PCR could be provided as supporting info
Can the author explain the meaning of activity scope and phenotype assessment?
The quality of the Tables is not good. It must be improved.
Toxicological analysis of drugs includes GC-MS, LC-HRMS. The authors should provide characteristic chromatograms and mass spectra data for each drug evaluation (retention time, calibration curves, conditions, analytes).
No specific comments on the quality of English language.
Author Response
Reviewer 2
We would like to thank the reviewer for his precious comment and for giving us the opportunity to revise our work. We appreciated his constructive comments on our manuscript, which led to a much-improved thorough revision of the manuscript. Our detailed answers – in bold - are interleaved to the reviews below.
Corrections and amendments made are also marked in yellow in the revised version of the manuscript.
R: “This study focuses on the evaluation of specific CYP450 enzymes, however the aim is not easy to understand. The authors should highlight mainly in the abstract and better in the introduction and in other parts the purpose of the study”.
A: The purpose of the study has been rewritten in the abstract, at the end of the introduction as well as in multiple parts of the discussion. As also suggested by Reviewer #1, and in other comments of Reviewer #2, the manuscript has been extensively rewritten in order to better clarify the aim and the concept of the paper. Thanks to the reviewers’ suggestions, we believe that the current version of the manuscript is much easier to understand.
R: “Some sections in Methods include Tables which could be moved in supporting info, parts 2.3 and 2.4 should be carefully written in a comprehensive manner. Results and discussion parts are not appropriate, a bit chaotic, without a specific row and must be reorganized. Perhaps the authors could present some specific targets for future studies.”
A: We moved Table 1 and 2 to the supplementary material as requested.
Parts 2.3 and 2.4 have been partially rewritten, in order to be more comprehensive and clearer, explaining the activity score and the phenotype assessment.
Results has been slightly modified, in order to be less chaotic and to include a lower number of subparagraphs. Moreover, wide parts of the discussion were rewritten or clarified and some specific targets for future studies have been suggested.
Tables have been modified for a better comprehension.
R: “The terms "g-phenotype" and "p-phenotype" should be explained better providing specific bibliography otherwise should be corrected.”
A: The terms g-phenotype and p-phenotype have been better explained at the end of section 2.4, rephrasing the sentence as follows: “Considering the terms used by Hicks et al. [1] to assign the activity prediction based on genotype test we propose the use of “g-phenotype”,that includes gPM, gIM, gNM and gUM.”, and “Following the suggestion of Shah et al. [7], we used the term “p-phenotype” to distinguish the phenotypic categories based on the true genotype (g-phenotype) from their phenotypic counterparts resulting from phenoconversion (p-phenotype).” Moreover, in the discussion we specified that ”On the basis of the genotyping and of the activity score-based assessment the phenotypic categories g-PM, g-IM, g-NM, and g-UM were identified and we adopted the term g-phenotype to group and to distinguish them from the p-phenotype term used to identify the phenotype after phenoconversion.”
We believe that in the current version the meaning of “g-phenotype” and “p-phenotype” are better clarified, and we thank the reviewer for this comment.
R: “Conclusion section seems like an opinion. I wonder if authors would address some major points and challenges of some demanding questions of the discussed area”.
A: following the reviewer comment, we modified the conclusions in order to address the major points and challenges demonstrated in our study related to the phenoconversion in the post-mortem setting. We further added some suggestions for future research.
R: “Detailed references should be provided for genotyped variants CYP2D6, CYP2C9, CYP2C19 and CYP2B6”.
A: we moved Table 1 in the Supplementary Material and added the respective references for genotyped variants.
R: “Primers for RT-PCR could be provided as supporting info”
A: We added the assay ID of all primers used for the CYP2D6 genotyping assay in Real Time PCR, as requested (paragraph 2.3). With this ID on the website of the manufacturer (ThermoFisher) the readership can trace the specifications of the product used.
R: “Can the author explain the meaning of activity scope and phenotype assessment?”
A: we thank the reviewer for this comment, which allowed us to better clarify some concepts. The meaning of activity score has been now added in section 2.4 and the translation from AS to phenotype has been better explained in the material and methods parts.
R: “The quality of the Tables is not good. It must be improved.”
A: we removed Tables 1 and 2, which have been moved to the Supplementary. We also introduced some modification in the Table 2 of the revised manuscript (corresponding to the Table 4 of the past version). Now tables are easy to read and to understand. To avoid misunderstandings, we suggest to the reviewer to visualize the tables in the “word version” of the manuscript. We noted that in the pdf version tables were not well formatted, and this was also highlighted to the editor.
R: “Toxicological analysis of drugs includes GC-MS, LC-HRMS. The authors should provide characteristic chromatograms and mass spectra data for each drug evaluation (retention time, calibration curves, conditions, analytes).”
A: We thank the Reviewer for his/her interest in the toxicological data. We better detailed the instrumentation in paragraph 2.2. Since all toxicological methods reported in the paper have been previously validated and published, we added references 19, 20, 21 and 22 where all the validation parameters (including retention times, calibration curves, conditions and analytes and other full validation parameters) are accurately detailed. We believe that, since the present study is not a validation paper, the readership could retrieve all information required in the “new” references, that were not specified in the original version of the manuscript.
Reviewer 3 Report
This manuscript presents a study on the genotyping of individuals post –mortem, toxicological analysis and determination of their potential phenotype concerning CYP2D6, 2C9, 2C19 and 2B6. In function of the toxicological detection of interfering drugs as inducers or inhibitors, the possible phenoconversion is determined by applying rules from the literature on the genotype.
The paper is well presented and clearly written.
Once you accept the phenoconversion rules applied by the authors and based on literature, you can follow the study.
The supplementary tables with all the individual finding is nice, since one can see exactly how the author have proceeded.
I will however make a few comments and critics
Table 1 : It would be worth also looking for CYP2C19*17 that seems to be relatively frequent and shows an ultra-metabolizer phenotype.
Table 2 : correct spelling : fenotype to phenotype.
Page 7 : table3 : If I understand well you calculate that a patient with an IM genotype is converted to a PM phenotype by simply multiplying the range by 0.5 when there is an inhibitor causing a drug-drug interaction.
This is strange? may be right, may be wrong, really depending of the case, the probe used for that CYP and many other parameters.
I see that you cite Borges in the Group of Pr Flockhart whom I respect. But I think this is a little too harzardous extrapolation.
For CYP2C9, Similarly you use inducers but there are also strong inhibitors (may be you did not have any in your patients).
From the paper you cite (ref 31), it looks that several group have used this notion of phenoconversion by simple calculation or application of a rule. Again this may be correct and evidently you cannot measure the real phenotype in dead people with partially proteolyzed organs contrary to the study in ref 38.. Thus only guessing can be done. For CYP 2B6 the reference given (38) compares actual phenotypes of microsomes, using S-mephenytoin as a probe (that can give a phenotype for CYP2B6 and 2C19 at the same time) with the genotype in a liver bank.
This extrapolation hurts my cartesian background.
Any how once you accept that the calculation rules are legitimate (since already applied in literature) then the paper is following those preceeding papers .
The experimental toxicological part (analysis) could described better with references to the methodology used.
I think that the paper could be accepted with minor corrections. Is “Metabolites” the correct journal?).
Author Response
We would like to thank the reviewer for his precious comment and for giving us the opportunity to revise our work. We appreciated his constructive comments on our manuscript, which led to a much-improved thorough revision of the manuscript. Our detailed answers – in blue and bold - are interleaved to the reviews below.
Corrections and amendments made are also marked in yellow in the revised version of the manuscript.
R: “Table 1 : It would be worth also looking for CYP2C19*17 that seems to be relatively frequent and shows an ultra-metabolizer phenotype.”
A: unfortunately, we have no possibility to analyze this additional CYP variant, out of our standard primers set, but we agree that it would be interesting also in forensic toxicogenetics and this is being already considered for future research.
R: “Table 2 : correct spelling : fenotype to phenotype.”
A: we corrected accordingly.
R: “Page 7 : table3 : If I understand well you calculate that a patient with an IM genotype is converted to a PM phenotype by simply multiplying the range by 0.5 when there is an inhibitor causing a drug-drug interaction. This is strange? may be right, may be wrong, really depending of the case, the probe used for that CYP and many other parameters. I see that you cite Borges in the Group of Pr Flockhart whom I respect. But I think this is a little too harzardous extrapolation.”
A: We used the works of Borges et al. [29] and Mostafa et al. [36] for the adjustment of AS score after phenoconversion, while Flockhart et al., as well as multiple other references (ref 11, 37, 38, 39), to classify substances and drugs as strong or weak inhibitors or as inducers. Clearly, our work strongly depends on the past published methods, but this has been nowadays quite well-established in the clinical setting, contrarily to the post-mortem one. On the other hand, we have reported in the manuscript that the identification of the influence of drugs on CYP enzymes, as well as their categorization as strong/weak inhibitors, remain difficult and this issue has been underlined in the revised version.
R: “For CYP2C9, Similarly you use inducers but there are also strong inhibitors (may be you did not have any in your patients).”
A: The phenoconversion of CYP2C9 was calculated according to the method proposed by Mostafa et al. [36], that does not consider the distinction between strong and weak inhibitors. When both inhibitors and inductors are present, the p-phenotype was not assessed. Table 2 was modified accordingly.
R: “From the paper you cite (ref 31), it looks that several group have used this notion of phenoconversion by simple calculation or application of a rule. Again this may be correct and evidently you cannot measure the real phenotype in dead people with partially proteolyzed organs contrary to the study in ref 38.. Thus only guessing can be done. For CYP 2B6 the reference given (38) compares actual phenotypes of microsomes, using S-mephenytoin as a probe (that can give a phenotype for CYP2B6 and 2C19 at the same time) with the genotype in a liver bank. This extrapolation hurts my cartesian background. Any how once you accept that the calculation rules are legitimate (since already applied in literature) then the paper is following those preceeding papers .”
A: We agree with the Reviewer regarding the fact that phenoconversion could only be guessed (followed the rules and scores as in ref 36) and we acknowledge the limitations and challenges represented by the evaluation of phenoconversion for CYP2B6 following ref. 42. Indeed, phenoconversion of CYP2B6 has been rarely applied. As also suggested by other reviewers, the discussion was extensively modified in order to better explain the scope and the concept of the study.
R: “The experimental toxicological part (analysis) could described better with references to the methodology used.”
A: We thank the Reviewer for his/her interest in the toxicological data. We better detailed the instrumentation in paragraph 2.2. Since all toxicological methods reported in the paper have been previously validated and published, we added references 19, 20, 21 and 22 where all the validation parameters (including retention times, calibration curves, conditions and analytes and other full validation parameters) are accurately detailed. We believe that, since the present study is not a validation paper, the readership could retrieve all information required in the “new” references, that were not specified in the original version of the manuscript.
Round 2
Reviewer 2 Report
I think the revised manuscript has been improved and is suitable for publication as a Protocol paper.
Minor editing of English language required.
i.e discussion section, the authors could use: .. In forensic practice, to understand CoD and MoD, toxicological results ..